Prioritizing riparian corridors for ecosystem restoration in urbanizing watersheds

Atkinson Samuel F. atkinson@unt.edu 1 2
Lake Matthew C. 1
1 Advanced Environmental Research Institute, University of North Texas , Denton , TX , United States of America
2 Department of Biological Sciences, University of North Texas , Denton , TX , United States of America
Jones Roger
Electronic publication date: 2020 Feb 4
Publication date: 2020
Volume: 8
Electronic Location ID: e8174
Received 2019 Jun 19; Accepted 2019 Nov 6
Copyright: ©2020 Atkinson and Lake
Copyright year: 2020
Copyright holder: Atkinson and Lake
License: This is an open access article distributed under the terms of the Creative Commons Attribution License, which permits unrestricted use, distribution, reproduction and adaptation in any medium and for any purpose provided that it is properly attributed. For attribution, the original author(s), title, publication source (PeerJ) and either DOI or URL of the article must be cited.
License URL: https://creativecommons.org/licenses/by/4.0/

Keywords: Ecosystem services, Water quality, Spatially explicit modeling, Watershed management, Water quality corridor management model, WQCM model

Funding: Upper Trinity Regional Water District (Texas) University of North Texas The work was supported by the Upper Trinity Regional Water District (Texas) and the University of North Texas. The funders had no role in study design, data collection and analysis, decision to publish, or preparation of the manuscript.

==============================
Background

Riparian corridors can affect nutrient, organic matter, and sediment transport, all of which shape water quality in streams and connected downstream waters. When functioning riparian corridors remain intact, they provide highly valued water quality ecosystem services. However, in rapidly urbanizing watersheds, riparian corridors are susceptible to development modifications that adversely affect those ecosystem services. Protecting high quality riparian corridors or restoring low quality corridors are widely advocated as watershed level water quality management options for protecting those ecosystem services. The two approaches, protection or restoration, should be viewed as complementary by watershed managers and provide a foundation for targeting highly functioning riparian corridors for protection or for identifying poorly functioning corridors for restoration. Ascertaining which strategy to use is often motivated by a specific ecosystem service, for example water quality, upon which watershed management is focused. We have previously reported on a spatially explicit model that focused on identifying riparian corridors that have specific characteristics that make them well suited for purposes of preservation and protection focused on water quality. Here we hypothesize that focusing on restoration, rather than protection, can be the basis for developing a watershed level strategy for improving water quality in urbanizing watersheds.

Methods

The model described here represents a geographic information system (GIS) based approach that utilizes riparian characteristics extracted from 40-meter wide corridors centered on streams and rivers. The model focuses on drinking water reservoir watersheds that can be analyzed at the sub-watershed level. Sub-watershed riparian data (vegetation, soil erodibility and surface slope) are scaled and weighted based on watershed management theories for water quality, and riparian restoration scores are assigned. Those scores are used to rank order riparian zones –the lower the score the higher the priority for riparian restoration.

Results

The model was applied to 90 sub-watersheds in the watershed of an important drinking water reservoir in north central Texas, USA. Results from this study area suggest that corridor scores were found to be most correlated to the amount of: forested vegetation, residential land use, soils in the highest erodibility class, and highest surface slope (r2 = 0.92, p < 0.0001). Scores allow watershed managers to rapidly focus on riparian corridors most in need of restoration. A beneficial feature of the model is that it also allows investigation of multiple scenarios of restoration strategies (e.g.,  revegetation, soil stabilization, flood plain leveling), giving watershed managers a tool to compare and contrast watershed level management plans.

Introduction

Many areas across the globe are rapidly urbanizing, resulting in land use and land cover changes that can adversely affect the natural resources upon which the growing population relies (Melchiorri et al., 2018). When urban growth is substantial, riparian corridors are often modified by human development activities. Modification of riparian corridors can alter vegetation, hydrology, channel morphology and water chemistry, including nutrients and pollutants (Walsh et al., 2005; Ramírez et al., 2012). In north central Texas the four counties that form the core of the Dallas-Fort Worth urban area (Collin, Dallas, Denton, and Tarrant Counties) are projected to grow to approximately 7.4 million by 2030, from its 2010 population of about 5.6 million, or a 30.9% increase in 20 years (US Bureau of the Census 2010 data, Texas Water Development Board 2030 projections). By 2070 the population is projected to increase to approximately 11 million. The watershed that was used as the test site for the study reported here, a 240,000 hectare watershed in the Dallas-Fort Worth area, has seen substantial growth in urban lands that accompanies the population growth in north central Texas. Lands classified as urban (i.e., residential, commercial, industrial and transportation) in this watershed have grown from 15,300 hectares in 2001 (6.4% of total land) to 29,800 hectares (12.4%) in 2014. These types of projections can be found for many large urban areas throughout the world. This growth puts pressure on all natural and infrastructure resources, including the freshwater resources that supply drinking water to these areas. Illegal refuse dumping, chemical runoff, clear-cutting, grazing, and other human influences are just a few of the detrimental factors affecting the dynamic balance between rivers and streams, their surrounding riparian corridors, and their encompassing catchments upon which these rapidly growing urban areas rely.

Riparian corridors

As one of the most diverse of habitat types, riparian corridors influence water quality, flood prevention, wildlife habitat, economics, and various other ecological, physical, biological, and chemical processes (Wagner, 2004). Riparian corridors reduce erosion (Castelle & Johnson, 2000), filter sediments and pollutants out of overland runoff (Staddon, Locke & Zablotowicz, 2001; Dabney, Moore & Locke, 2006; Bongard et al., 2010), block solar radiation to moderate water temperature (Sweeney, 1992), provide habitat (Jones III et al., 1999), and store water thereby moderate flooding (Cooper, Hiscock & Lovett, 2019). However, on a global basis more than 32,000 large dams (defined by the International Commission on Large Dams as a dam with a height of 15 m or greater, or a dam between 5 m and 15 m impounding more than 3 million cubic meters (ICOLD, 2011)), and upwards of 1,000,000 total dams have fragmented systems of streams and rivers (Jackson et al., 2001), leading Dudgeon et al. (2006) to state that freshwater ecosystems “…may well be the most endangered ecosystems in the world”. The US Geological Survey’s John Wesley Powell Center for Analysis and Synthesis further suggests that “fresh water is arguably the most valuable resource on the planet, but human activities threaten freshwater ecosystems” (USGS, 2019). Effective assessment and management techniques are necessary to protect the diversity of the ecosystem services found within these ecosystems, and to mitigate current and future conditions of environmental stressors amplified by rapid urban development.

Scientists have long studied riparian corridors as ecosystems of forested and/or vegetative transition zones that link aquatic and terrestrial environments (see, for example Karr & Schlosser (1978). Studies show that healthy riparian stream corridors perform a multitude of valuable services for their adjacent waterways, including: their overall influence on water quality (e.g., Connolly et al., 2015); biological diversity (e.g., Mlambo et al., 2015); ecosystem maintenance (e.g.,  Montgomery, 2001); and protection of intermittent streams and the residual pools that they provide as refuges for multiple species during dry periods (e.g., Wigington Jr et al., 2006). Nutrient cycling, contaminant filtration, water purification, bank stabilization, stream temperature maintenance, flow stabilization, flood attenuation, and habitat preservation are some of the numerous functions carried out by riparian zones (National Research Council, 2002). In a recent review of riparian restoration literature (Feld et al., 2018), it was stated that riparian restoration provides “a no-regrets management option to improve and sustain lotic ecosystem functioning and biodiversity”.

Riparian corridors exhibit a wide variety of geomorphic characteristics across a continuum of scales. Studies that need to define the lateral extent of those corridors for research purposes have used a range of approaches, from defining each riparian corridor on an individualized case-by-case basis, to defining all corridors simultaneously in a study area based upon more generalized definitional basis. An example that is nearer the individualized end of the spectrum, a recent study that developed a model to measure departures of current riparian conditions from historic conditions (defined as pre-European settlement in the north-western United States) used a stream segment’s valley bottom width as the lateral riparian extent because that represents the maximum possible extent of riparian vegetation (Macfarlane et al., 2017). Using Thiessen polygons with centroids located at the midpoint of each stream segment, they defined the individual lateral extent of each steam segment’s riparian corridor across an irregular planform of geometries and valley bottoms. Nearer the other end of the spectrum, Gilbert, Macfarlane & Wheaton (2016) developed a tool that set the maximum valley bottom (defined as the stream or river channel and the associated low-lying floodplain) as the maximum possible extent of a riparian area. That tool set the lateral extent of riparian areas for a given size of a drainage area (classified as either “confined”, “transition” or “large” based on km2) based on a buffer around the stream channel set to the maximum valley width in the drainage area.

Watershed managers concerned with water quality find that suggesting restrictions on development in high quality riparian corridors, or restoring degraded riparian corridors, is particularly challenging in Texas. Water law in Texas has evolved from conflicts between private landowners and the needs of Texas citizens. Privately owned lands dominate the landscape of Texas; less than 2% are federal lands (Vincent, Hanson & Argueta, 2017). Additionally, two legal doctrines of surface water law are recognized in Texas today: the riparian doctrine, and the prior appropriation doctrine (Sansom, 2008). The fundamental concept of prior appropriation is based on pre-statehood Spanish law that evolved to control water use in irrigation intensive settlements in Texas and the southwest. Those Spanish law concepts have today evolved such that Texas owns most surface water, holding it in trust for the public, and allocates use of that resource through a permitting process. Permits are issued on a “first in time, first in right” for a given allocation of water. The riparian doctrine is based on an older English common law approach to water use. The basic concept is that private water rights are tied to the ownership of the land bordering a natural river or stream. Thus, surface water rights in Texas are tied to both state permitted appropriations and by land ownership (Wurbs, 2004), effectively restricting watershed managers’ options, and for all practical purposes limits the lateral extent of a riparian corridor that a watershed manager can consider for water quality purposes.

Here, we offer a strategy for prioritizing riparian corridors for potential ecosystem restoration focused on water quality. Fortunately, efforts focused on a specific ecosystem service can also enhance other ecosystem service. For example, Fremier et al. (2015) focus on riparian restoration as a means for improving connectivity between protected habitats as a means to increase the ecological resilience of those protected habitats. Their work was motivated in part much the same as ours; increasing a system’s ability to respond to natural and human induced perturbations. Although their focus was for enhancing biodiversity while ours was for improving water quality, the net result of either focus would likely yield benefits to the other. Other beneficial synergistic restoration outcomes could include aesthetics, in-stream habitat, fish passage and bank stabilization (see, e.g., Bernhardt et al., 2005).

Effects of urbanization

Urbanization and its subsequent activities, without proper planning, often leads to the degradation of streams and their riparian corridors. This degradation may affect the natural cycles of biological and physical activities normally carried out within riparian ecosystems (Tanaka et al., 2016), and also cause social and economic problems at both local and regional levels. For example, Withers & Jarvie (2008) state that the problems can influence human health (e.g., algal toxins), species abundance and diversity, amenity value and costs of water treatment for drinking. Streams and rivers on the periphery of urbanizing areas are particularly vulnerable due to population density, sensitivity to land use change and ubiquitous exploitation (Walsh et al., 2005). Furthermore, many of those aquatic systems at the urban periphery have already been exposed to agricultural practices, such as grazing and the direct access of cattle to streams. This has resulted in increased erosion of stream banks due to trampling, as well as direct deposition and indirect flow of animal waste into waterways, a principal component of nonpoint source pollution (Hamilton & Miller, 2002). Ultimately, the dynamic equilibrium of stream ecosystems can be altered by the cumulative effects of channelization, clear-cutting, illegal dumping, and increased chemical usage, all consequences of urbanization of surrounding riparian corridors.

Many places throughout the world are experiencing the same pattern of land use change that can be seen in north central Texas, the application site for this model. Former rural areas are becoming a part of an ever increasing urban landscape. As residential developments, commercial properties, and industrial facilities increase, they cover the natural landscape with roads, buildings, parking lots, and other impervious surfaces (Wang et al., 2019). Stream health can be directly linked to urbanization, the effects of which simultaneously decrease bank stability and increase pollutant presence and transfer. Healthy riparian buffer zones have been shown to filter out up to 97% of soil sediment prior to stream entrance (Lee, Isenhart & Schultz, 2003). However, removing trees in riparian zones is often one of the first activities associated with urban and suburban development, leading to increased soil erosion. Increased erodibility results in a decrease in the depth of fertile topsoil and an increase of sediment within streams. These sediments often contain metals such as lead, chromium or zinc, pesticides such as DDT, other organics such as polychlorinated biphenyls (PCBs) or polycyclic aromatic hydrocarbons (PAHs), or various other synthetic chemicals that may be toxic to aquatic and terrestrial species, and may also be linked to human health via the food chain (see, for example: Pavlović et al., 2016; Walters et al., 2018; Bing et al., 2019).

The 2000 National Water Quality Inventory (US Environmental Protection Agency, 2000) indicated that 39% of the river and stream miles in the United States were listed as impaired or polluted, but the updated 2017 report indicated that the number increased to 46% (US Environmental Protection Agency, 2017). River and stream conditions were assessed with measures of biological quality, chemical stress, physical habitat stress, or human health indicators. Aquatic conditions have led to extinction rates of freshwater fauna that are five times that for terrestrial biota (Ricciardi & Rasmussen, 1999; Sand-Jensen, 2001). Fortunately, aquatic ecosystem restoration can lead to improved biological, chemical and physical conditions, resulting in both improved wildlife habitat and water quality.

Objectives of this study

In an earlier study focused on prioritizing and protecting the highest quality riparian corridors, a spatially explicit modeling and mapping technique was developed; the Water Quality Corridor Management (WQCM) model (Atkinson et al., 2007; Atkinson, Hunter & English, 2010), originally referred to as the “wick ‘em” model because of the acronym. We now call that model the WQCM-P model, because it is focused on “protection” strategies. The model involves a geospatial database that utilizes GIS and remote sensing techniques to assess and prioritize stream reaches according to their overall health and sustainability. That model was developed and then assessed for its ability to document stream corridor quality, and ultimately, establish a ranking system for developing management strategies for protecting the highest quality stream corridors for drinking water quality purposes.

The model’s application area for this research consists of an important watershed in north central Texas, USA. The overall watershed is 240,000 hectares in size (593,000 acres) and drains to a relatively large drinking water reservoir (12,000 hectares; 29,600 acres), Lewisville Lake, which serves the Dallas /Ft. Worth area (current population approximately 6 million). The watershed was divided into 90 sub-watersheds based on US Geological Survey Hydrologic Unit Codes, ranging in size from 230 to 12,000 hectares (570 to 29,000 acres). Atkinson, Hunter & English (2010) provides a map that shows the results of the original WQCM-P modeling applied to that watershed. That map illustrates to watershed managers which sub-watersheds contain the highest priority stream corridors that should receive the initial focus for an overall watershed riparian protection program.

Because high quality riparian corridor protection is only one strategy available for water quality purposes, it became apparent after the release of the WQCM-P model that a tool was also needed to rank riparian corridors in terms of potential aquatic ecosystem restoration. A re-working of the original model was hypothesized to potentially provide that new focus. The new work described in this paper turns the riparian protection question around, and asks the model to prioritize riparian corridors for potential restoration activities in order to improve water quality in the reservoir. The objective of the research reported here was to modify the original model, using the same spatial databases, to one focused on potential “restoration” opportunities, or the WQCM-R model. Using the same databases to reframe the question asked in this research allowed an analysis of the effects of the reframing of the question, without a confounding effect of also changing databases. Therefore, the new model described in this paper used the same sub-watershed delineations, stream network, land use, soils, and topography data as was used the original model.

A subsequent benefit of this new model is that not only will it indicate which sub-watersheds contain riparian corridors that have the greatest restoration needs, it also allows targeting of sub-watersheds that would receive the largest increases in riparian corridor quality based on specified restoration strategies (e.g., revegetation with native plants, soil stabilization, etc.). As Nilsson et al. (2015) state, natural river ecosystems are both self-sustaining and dynamic, one of their stated goals for successful river restoration. Our goal is that WQCM-R will assist watershed managers in choosing not only the riparian areas most in need of restoration, but also help them identify the most appropriate restoration strategy. Using both the WQCM-R and the WQCM-P models, watershed managers should have access to a broad range of riparian corridor options for assisting in water quality considerations in rapidly urbanizing watersheds.

Materials & Methods

The WQCM-R model, reported on here, was designed to (1) utilize easily accessible data for the purpose of identifying and assessing potential water quality issues and (2) to classify stream segments in order of riparian quality in order to prioritize potential restoration activities as a component of an overall watershed management plan. The model generates scores for each sub-watershed based on riparian zone characteristics for each of the spatial variables considered by the model—lower scores imply higher restoration priority. While results are listed for each sub-watershed for purposes of clarity, in actuality the results are limited to data extracted from only the corridors.

GIS materials

The following spatial data layers were utilized in the generation of the WQCM-R model:

Sub-watershed and its Corridor Component: US Geological Survey (USGS) Hydrologic Unit Codes were used to define 90 sub-watersheds in the overall Lewisville Lake drainage. These sub-watersheds ranged in size from 230 to 12,000 hectares (570 to 29,000 acres). ESRI ArcInfo’s buffer tool was used to generate a 20 meter (66-foot) wide buffer zone around a stream shapefile (40 meter total width) obtained from the USGS National Hydrography Dataset (NHD1), which is provided at a scale of 1:24,000. The stream buffer was then exported as a separate shapefile to be used as the extent from which the land use, soil erodibility, and surface slope were clipped. Because sub-watershed varied substantially in area, all subsequent data were transformed with a scaling factor: the ratio of total riparian area within a sub-watershed to total sub-watershed area.

Land Use Parameter: Using the same land use classifications from our previous modeling effort (Atkinson et al., 2007; Atkinson, Hunter & English, 2010), surface area of eight classes of land use were extracted for each riparian corridor in the study area: barren, cropland/pasture, forested, residential, shrub/brush rangeland, urban, water, and unclassified. The land use classes were derived from 30-m resolution LANDSAT 8 ETM satellite imagery using Definiens eCognition object-based image classification software. The native scale of 30 m raster cells were utilized for all raster data.

Soil Erodibility Parameter: Using the same erodibility data from our previous model, the Soil Survey Geographic (SSURGO) data from the National Resource Conservation Service was used to create a soil erosion potential shapefile. There were eight erodibility ‘kffact’ categories in the study area, where higher numbers indicate higher potential erosion: Kw = 0, Kw = 0.17, Kw = 0.20, Kw = 0.24, Kw = 0.28, Kw = 0.32, Kw = 0.37, and Kw = 0.43. Soil erodibility classes were extracted for each riparian corridor in the study area.

Surface Slope Parameter: The same Digital Elevation Model (DEM) from our previous model was used for developing surface slope in the study area. Thirty meter raster data were obtained from the USGS National Elevation Dataset (NED) to create a percent slope raster file. Those data were categorized into five potential surface slope categories: <1%, 1% to <2%, 2% to <3%, 3% to <4%, and 4% to 5% (no riparian corridors had areas with slopes greater than 5%).

Methods used in original water quality corridor model for protection –WQCM-P

The original WQCM-P model utilized the same four geospatial variables listed above, but also included a “floodplain parameter” which accounted for the percent of a riparian zone that fell within a Federal Emergency Management Agency (FEMA) 100-year floodplain. This was included in the WQCM-P model because it helped account for some “protective” measures already in place (e.g., restrictions on certain types of developments within the floodplain).

Each of the spatial variables consists of an importance weight and a scaling function (details provided in Atkinson et al. (2007) and (Atkinson, Hunter & English, 2010)). Importance weights and scaling functions assigned to each variable range from 1 to 5, with 5 indicating a greater need for protection. For example, land use class was considered the most important variable in the model, receiving 5 importance points, and the scaling function for land use indicates that forested areas within the riparian buffer receive 5 points while residential areas within the riparian buffer receive 2 points. Each variable’s scaling function was based on the same concept: what conditions are more relevant to protect via preservation? Two of the variables are specific to preservation goals. First, the FEMA 100 year Floodplain variable was considered because areas designated to be inside these floodplains already receive some amount of preservation protection (e.g., certain types of activities require flood insurance which often discourages that activity). Second, the Corridor Ratio variable is the ratio of stream’s corridor area to its sub-watershed area. The Corridor Ratio and was considered because larger corridor ratios suggest less room for development in the sub-watershed and therefore more pressure to develop inside a stream’s corridor area. In the WQCM-P model, higher scores represent higher quality and therefore should be targeted for riparian corridor protection.

Changes in methods for the water quality corridor model for restoration—WQCM-R

The newly developed WQCM-R, established for restoration potential modeling, eliminated the floodplain variable and the corridor ratio variables used in WQCM-P because they were specifically focused on preservation strategies. Because floodplain development restrictions typically do not affect “restoration” of riparian zones, these parameters were eliminated from the WQCM-R model. The remaining four variables were retained for WQCM-R but importance and scaling functions were altered to reflect a focus on restoration potential. Final scores were calculated for each corridor in each sub-watershed’s riparian area based on characteristics of three variables: land use (L), erodibility (E) and slope (S). Values were calculated for the forth variable, the area of the 20 m wide corridor on each side of the stream segments in the study area, generating an overall WQCM-R score for the riparian corridors of each sub-watershed.

The underpinning model is presented in Eq. (1): (1) WQCM-R Score=LiLf+EiEf+SiSf

Where the subscripts “i” and “f” represent “importance” and “functional scale” (Table 1).

Table 1 Importance and weighting for each variable in WQCM-R model.

WQCM-R Variable		
Land Use (L)Importance= 5	Eight classes were generated. Native vegetative cover (forested riparian zones) within the stream corridor, are considered to have higher quality for drinking water purposes.	
Class	Class importance	
f = forest	5	
w = water	5	
s = shrub/brush	4	
c = crop/pasture	3	
b = urban	2	
u = residential	2	
r = barren	1	
u = unclassified	5	
Lf= (cfaf + cwaw + csas + ccac + ccab + cuau + crar + cuau) / SWa Where Lf is the land use function, ci is class importance, “ai” represents area in acres of class ci, and SWa is the sub-watershed area.	
Erodibility (E)
Importance= 3	Erodibility (Kw) in study area ranged from 0 to 0.43; lower Kw soils have less potential for erosion and are considered to have higher quality for drinking water quality purposes	
Class	Class importance	
Kw = 0.00	5	
Kw = 0.17	5	
Kw = 0.20	4	
Kw = 0.24	4	
Kw = 0.28	3	
Kw = 0.32	3	
Kw = 0.37	2	
Kw = 0.43	1	
Ef= (cfaf + cwaw + csas + ccac + ccab + cuau + crar + cuau)/SWa
Where Ef is the erodibility function, ci is class importance, “ai” represents area in acres of class ci, and SWa is the sub-watershed area.	
Slope (S)
Importance= 2	Slope range from <1% to 5%; Gentler slopes soils have less potential for erosion and are considered to have higher quality for drinking water quality purposes.	
Class	Class importance	
slope <1%	5	
slope = 1% to <2%	4	
slope = 2% to <2%	3	
slope = 3% to <4%	2	
slope = 4% to 5%	1	
Sf= (cf af + cwaw + csas + ccac + ccab + cuau + crar + cuau) / SWa
Where Sf is the slope function, ci is class importance, “ai” represents area in acres of class ci, and SWa is the sub-watershed area.	

WQCM-R scores were generated for each sub-watershed’s riparian corridors based on Eq. (1). Each riparian corridor’s score was associated and mapped onto the appropriate sub-watershed, and all sub-watersheds were then placed into one of four priority quartiles: low, moderate, high, and highest restoration priority. Scores can theoretically range from a low of 10 (a sub-watershed whose riparian zones are comprised of 100% barren land, 100% highly erodible soil, and 100% steep surface slope), to a high of 50 (riparian zones comprised of 100% forest or water, 100% low erodibility soil, and 100% negligible surface slope). This model intends watershed managers to focus riparian restoration efforts on lower scores, those that represent the best opportunities for watershed level water quality management.

Statistical analyses

Principal component analyses. Data for 20 variables characterized each sub-watershed’s riparian corridor: area of each of 7 land use classes; area of each of 8 soil erodibility classes, and; area of each of 5 surface slope classes. Principal component analyses were used to examine the sources of variation in the 20 variables and determine how many principal components were needed to explain at least 50% of the total variation, as well as determine if the variables that formed the largest eigenvectors of the first principal component corresponded with the variables that formed the best multivariate linear regression model.

Least squares regression. Least squares multivariate linear regression analysis was also applied to the WQCM-R results, seeking the best regression model using a maximum r-squared approach. Regression modeling set a sub-watershed’s riparian area WQCM-R score as the dependent variable, and each of the 20 riparian characteristics for the sub-watershed’s riparian area as the independent variables. The criteria for “best” included:

1. each variable must be logically consistent in terms of the sign of its coefficient

2. each variable must be statistically significant

3. the addition of each variable must increase the adjusted r2 by a minimum of 0.05

4. the overall model must be statistically significant.

Results

Each sub-watershed’s environmental data were entered into a GIS database, and the corresponding 20-meter wide riparian corridor data were extracted. For each corridor up to seven land use classes were identified and mapped, up to eight soil erodibility classes were mapped, and up to five surface slope classes were mapped. Due to varying sizes of the sub-watersheds and their riparian corridors, each mapped class was also normalized by the ratio of riparian area to sub-watershed area to allow comparisons across all sub-watersheds. Normalization allows comparisons to be based on the percent of each environmental class in a riparian zone as opposed to the absolute area of each class.

Principal component analyses of the normalized data (20 variables) found that the first three principal components explained more than 50% of the variance of the data (Fig. 1). The five largest eigenvectors of the first principal component were derived from: surface slope between 4% and 5%, Kw28 and Kw32 erodibility soils, forested vegetation and surface slope less than 1%.

Figure 1 Principal components (90 observations, 20 variables): first three principal components explains 54.3% of total variance.

Proportion of total variance explained by each eigenvalue (solid line) and cumulative proportion of total variance (dashed line).

The WQCM-R algorithm was next applied to the riparian zones in each of the 90 sub-watersheds in the study area. Based on the resultant score, each sub-watershed was placed into a corresponding WQCM-R restoration priority quartile (low, moderate, high, or highest). Figure 2 provides a restoration priority map, where sub-watersheds with the darkest color represents the highest priority (lowest scores) for stream corridor restoration. This mapping technique allows watershed managers to visualize where stream corridor restoration activities will generally be most beneficial based on a sub-watershed’s quartile, and more specifically where to target restoration activities within the smaller riparian area of the sub-watershed.

Figure 2 WQCM-R: priority for riparian restoration.

Darkest color represents highest priority for restoration quartile, lightest color represents lowest priority quartile for restoration.

Least squares linear regression analysis was also applied to the WQCM-R results, seeking the best regression model using a maximum r-squared approach.

Using the criteria established to define the best multivariate regression model, the best model identified included four variables (r2 = 0.922, adjusted r2 = 0.919, p < 0.0001): percent of riparian area with forested vegetation, percent of riparian area with residential land, percent of riparian area with soils in the highest erodibility class, and percent of riparian area with highest surface slope class. As would be expected for an algorithm that yields lower scores for areas more in need of restoration, Table 2 shows that forested land has a positive coefficient (increasing forest area results in higher WQCM-R scores and less potential need for restoration), while residential land, high erodibility soil and high surface slope all have negative coefficients (increasing area results in lower WQCM-R scores and more potential need for restoration).

Table 2 Regression model variables coefficients.

WQCM-R score is dependent variable; independent variables found to best predict WQCM-R score in a riparian corridor were forested area, residential area, highest erodibility soil area and highest surface slope area are independent variables.

“Best” Regression Model to Predict WQCM-R score	Intercept	Independent Variables	
		Amount of Corridor with Forest	Amount of Corridor with Residential	Amount of Corridor with Highest Erodibility Soil	Amount of Corridor with Highest Surface Slope	
Parameter Estimate	37.47442	6.01232	−10.60458	−5.51017	−8.55126	
F-Value	99,413.5	321.94	123.85	122.66	619.15	
Pr > F	<0.0001	<0.0001	<0.0001	<0.0001	<0.0001	
r2 = 0.9224, p < 0.0001, Adjusted r2 = 0.9187	

Figure 3 illustrates a comparison of predicted restoration scores based on the best four variable linear regression modeling versus the twenty variable WQCM-R scores. The data in the Fig. 3 represent the 90 sub-watersheds sorted from highest to lowest WQCM-R restoration priority, illustrating the close correlation between the twenty variable weighted-scaled WQCM-R approach and the four variable regression modeling approach, suggesting the simpler approach may be preferred. However, the WQCM-R approach allows exploration of multiple restoration strategy scenarios, as will be described below, which the regression modeling approach does not.

Figure 3 Multivariate regression prediction of restoration priority compared to WQCM-R modeled restoration priority for 90 sub-watersheds, ordered from highest to lowest WQCM-R restoration priority (left to right).

Solid points represent WQCM-R modeled priority for a sub-watershed and open points represent multivariate regression prediction for the same sub-watershed.

A valuable aspect of the WQCM-R model, as opposed to the simpler regression model, is that in addition to an overall restoration potential score, specific landuse, soil erodibility, and surface slope data are also available for each corridor. This additional information allows watershed managers to glean insight as to why and how a particular corridor should be restored, and explore the overall watershed implications for various restoration strategies. For example, if a restoration strategy was focused on re-vegetation of crop/pasture land in a riparian zone with native tree species, it is straightforward to determine how effective that strategy would be in terms of improving overall riparian scores. It is a simple matter within the model to reclassify crop and pasture lands within riparian corridors to forested land and recalculate WQCM-R scores. For the 90 sub-watersheds in this analysis, this restoration strategy (re-vegetation of crop/pasture lands with native trees) would result in an increase of average WQCM-R score from 36.68 to 37.11 (1.18%), a positive outcome. The resultant change is shown in Fig. 4, where the 90 sub-watersheds are ordered from lowest to highest original score on the x-axis and displayed with a solid black line (lower scores have higher restoration priority). New scores for a given riparian corridor based on a revegetation restoration strategy are shown with green data points. As can be seen in the graph however, very little of the improvement would occur in the riparian corridors with the lowest original scores, the sub-watersheds most in need of restoration. This information would be very useful for a watershed manager whose intent is to focus on improving the lowest scoring sub-watersheds.

Figure 4 Improvement in WQCM-R scores based on two different restoration strategies (restore native riparian vegetation or stabilize soils).

The black line represents original restoration priority score, green points represent improvement in score under a re-vegetation restoration strategy and orange points represent improvement in score under a soil-stabilization restoration strategy.

A second restoration strategy might be to focus on stabilizing the most erodible soils using the same approach as above, and reclassifying highly erodible soils into a lower Kw class to represent soil stabilization. A restoration scenario focused on the most erodible soils would increase scores from 36.68 to 37.63 (2.60%). Figure 4 represents this scenario with orange data points, and it is clear that much of the improvement would occur in sub-watersheds in the lowest quartile scores. A watershed manager might prefer this strategy because it is more closely aligned with the conceptual design of the WQCM-R model—improving the most problematic riparian corridors first. This type of scenario analysis is a useful benefit of a spatially-based riparian restoration model such as WQCM-R.

Discussion

WQCM-R was developed to be generally applicable to any rapidly urbanizing watershed. The model was applied to a major drinking water reservoir in north central Texas, because the region is one of the fastest growing regions in the country. Of the top 6 fastest growth rates in large cities between 2017 and 2018, two are in this region: Frisco and McKinney, and Fort Worth experienced the third largest overall population increase in the United States during the same time (US Census Bureau, 2019). Rapid urbanization in the north central Texas area has been occurring for several decades. An analysis of the US Department of Agriculture/US Department of Interior’s LANDFIRE data base for north Texas land use indicates the degree of urbanization in the sub-watersheds in the study area indicates a substantial increase in urbanization between 2001 and 2014 (US Geological Survey, 2014). Of the 90 sub-watersheds analyzed, 32 experienced more than 10% increases in urban lands during that time, five of which increased by over 100%. Of the 90 associated riparian corridors, exactly 30 had increases in urban lands of over 10%, four of which experienced more than 100% increases of urbanization. Figure 5 shows the relationship between the percent change in urban lands of a riparian corridor and the percent change in urban lands in the associated sub-watershed. This relationship (r2 = 0.83, p < 0.01) indicates why watershed planners are concerned with urbanization pressures on riparian corridors—urban development in a sub-watershed will almost certainly encroach on nearby riparian corridors.

Figure 5 Percent change in urban lands in sub-watershed versus percent change in urban lands in the associated riparian corridors.

There is a high and statistically significant relationship between percent increase in urban land use in a sub-watershed and the percent increase in urban land use in the sub-watershed’s riparian corridors.

A visual examination of urban growth in the northern Dallas/Fort Worth metropolitan area over the past 3 decades is also revealing. Figure 6 shows three snapshots in time of satellite imagery of the area, spanning 32 years, (A) a graphic that labels dominant features in the imagery; (B) 1984; (C) 2000; (D) 2016. Urban land use is quite distinct (brighter pixel values seen predominantly in the southeast sectors of the imagery) and the expansion of urban land use is obvious. In 1984 Lewisville Lake (in center of satellite images) has some urban lands immediately to the southwest, and some to the northwest. Ray Roberts Lake (top center) did not exist in 1984, and Grapevine Lake (lower left corner) had very little urban land immediately surrounding the lake. DFW International Airport, one of the more dominant features in the imagery (bottom center) was completed in 1974, and by 1984 there is only moderate amounts of urban land use near the airport. However, by 2000 there is substantial urban development around the airport and noticeable urbanization around both Grapevine and Lewisville Lakes, and Ray Roberts Lake has been completed, but there is little urban land use around the new lake. By 2016, urbanization has almost completely surrounded both Grapevine and Lewisville Lakes, and there are hints of urban land use near Ray Roberts Lake. This trend is likely to continue, with more and more urban development occurring as the Dallas Fort/Worth metropolitan area continues its rapid growth to the north. The Lewisville Lake watershed has experienced a high degree of urbanization in the southern half of its watershed over these 3 decades, and the northern portions are likely to urbanize if pressure for development around Ray Roberts Lake follows a similar pattern as that which occurred around Grapevine and Lewisville Lakes.

Figure 6 Urbanization in north central Texas observed from satellite imagery.

(A) represents major landmarks in the imagery for orientation; (B) satellite imagery acquired in 1984; (C) satellite imagery acquired in 2000; (D) satellite imagery acquired in 2016. Imagery shows rapid urban development (urban land use tends to be brighter that natural vegetation and agricultural lands). The expansion of brighter pixels from the lower right (southwest) corner of the images up and to the left coincides with population growth and coincident water demand in the Dallas/Fort Worth region. ©2018 Google Earth: Images Landsat/Copernicus.

The rapid urbanization of Dallas/Fort Worth region, along with its associated population growth, has alerted watershed planners of the need to focus on potential ways to safeguard water quality in the sources of drinking water for more than 6 million people currently living in the area. One of the first things these planners considered was targeting riparian corridors that were highly functioning in terms of water quality services for potential protection from development pressures that tend to reduce water quality services. This initially led to the development of the WQCM-P model that has been described above, which was applied initially to the Lewisville Lake watershed. That effort ranked 90 sub-watersheds in terms of priority for protective actions (e.g., a Water Authority might purchase corridors, might offer incentives to land owners for limiting development). Next, watershed managers wanted to consider targeting riparian corridors that were poorly functioning in terms of water quality for potential restoration projects that improve water quality services.

Initially, it seemed reasonable to assume that riparian corridors which were ranked low for protection actions would be those that were most appropriate for restoration. However, multiple discussions with local watershed managers lead to the realization that two variables used in WQCM-P to identify riparian corridors for protection, FEMA floodplain designation and riparian area to sub-watershed area, were not the most appropriate considerations for identifying corridors for restoration. Additionally, the importance weighting and scaling that had been applied to riparian corridor variables under protection strategies needed adjustment under restoration strategies. After the development of the WQCM-R model, a rational question to explore was whether WQCM-P scores and WQCM-R scores were highly correlated. Conceptually, there should be a negative relationship: the higher the potential for protection based on water quality services, the lower the need for restoration. If they were highly negatively correlated, the WQCM-R would not be necessary, and watershed managers could simply target riparian corridors that had low scores for protection as those that had high potential for restoration. Figure 7 shows the relationship between the WQCM-P and the WQCM-R scores for the 90 sub-watersheds under review. The data show a negative correlation, but the variation in WQCM-P that explains WQCM-R is very low ( r2 = 0.052). However, the trend is statistically significant (p = 0.03) confirming the conceptual assumption that there is a statistically significant negative relation between the protection and the restoration models, but not one that can be used to predict the other.

Figure 7 WQCM-P scores versus WQCM-R scores.

Regression line shows a statistically significant negative correlation between protection/preservation priority and restoration priority, as would be expected, but the coefficient of determination suggests that very little of the variability in WQCM-R can be explained by WQCM-P alone (r2 = 0.052, p = 0.03).

WQCM-R priority scores of each sub-watershed’s riparian corridors are the primary tool that watershed managers use to develop a targeted restoration strategy. For visual simplicity, a map is produced that divides the sub-watershed into restoration priority quartiles (for example, see the WQCM-R sub-watershed quartile map above), and is the product that guides a watershed manager to find actual corridors suitable for restoration. In practice, a watershed manager will examine each sub-watershed in the highest priority quartile, using the detail of high resolution aerial imagery to locate candidate restoration opportunities. For example, in the Lewisville Lake watershed, Fig. 8 shows (A) an orientation map that depicts the location of a high restoration potential sub-watershed and (B) a corresponding 2106 National Agriculture Imagery Program (NAIP) image with the 40 meter wide riparian corridor boundary highlighted. Figure 9 shows (A) an orientation map that depicts the location of a low restoration potential sub-watershed and (B) a corresponding 2016 NAIP image with its 40 meter wide riparian corridor boundary highlighted. These two examples represent how WQCM-R would be used in practice; the quartiles map focuses attention on where the watershed manager should look more closely to identify restoration opportunities.

Figure 8 Visual examination of portions of a riparian corridor in a sub-watershed within the highest riparian corridor restoration quartile.

(A) Orientation map that highlights which highest priority sub-watershed is considered and location of the NAIP imagery within the sub-watershed. (B) The corresponding 2016 NAIP imagery with the 40 m riparian buffer highlighted. Public domain imagery: US Department of Agriculture Farm Service Agency: National Agriculture Imagery Program, 2016.

Figure 9 Visual examination of portions of a riparian corridor in a sub-watershed within the lowest riparian corridor restoration quartile.

(A) Orientation map that highlights which lowest priority sub-watershed is considered and location of the NAIP imagery within the sub-watershed. (B) The corresponding 2016 NAIP imagery with the 40 m riparian buffer highlighted. Public domain imagery: US Department of Agriculture Farm Service Agency: National Agriculture Imagery Program, 2016.

Conclusions

It has been estimated that riparian ecosystems once covered 75 to 100 million acres (30 to 40 million hectares) in the contiguous United States, and that by the mid 1980’s two-thirds of that land had been converted to other non-native land-uses (Swift, 1984). Recognition of the loss of ecosystem services accompanying those changes, for example the loss of water quality protection, led to conservation efforts focused on riparian areas. Because it is typically less expensive to preclude conversion of existing native riparian areas than it is to restore riparian areas that have already been converted, initial conservation efforts normally focus on preservation. However, riparian corridor restoration, especially in rapidly urbanizing watersheds, is likely to become more prevalent as water quality managers seek watershed based management approaches to protect and enhance water quality. This is becoming more evident in light of increasing benefit/cost studies that show the value of restored riparian ecosystem services exceeds the costs of those restoration efforts (e.g., Holmes et al., 2004; Benayas et al., 2009; Costanza et al., 2014; Daigneault, Eppink & Lee, 2017; Uggeldahl & Olsen, 2019), even though restoration is typically more costly than preservation.

Based on the approach utilized in the development of the original WQCM-P model to identify high quality riparian systems for potential preservation purposes, it was hypothesized that the model could be re-worked into a different tool that could be used to prioritize riparian corridors in terms of potential riparian restoration. The new goal was to develop WQCM-R to assist watershed managers in rank ordering riparian areas most in need of restoration from a water quality ecosystem service perspective, and to also help them identify the most appropriate restoration strategy for those riparian corridors. It was conceived of as being a straightforward and uncomplicated tool that could be applied to urbanizing watersheds that drain into drinking water reservoirs, allowing prioritization of the sub-watersheds in terms of riparian corridor restoration potential. An additional benefit of WQCM-R would be that it allows exploration of multiple riparian restoration strategies prior to expending resources to implement a watershed management plan.

The development of WQCM-R was based on watershed management theories that suggest which environmental variables should be included in an examination of riparian zones, as well as how important each variable would be for prioritizing a riparian zone for potential restoration for water quality purposes. The model was developed based on commonly available information and applied to an important watershed in north central Texas, USA. The results confirmed that high surface slope, high soil erodibility and lack of forested vegetation (in that order) were the driving factors in restoration potential. Because WQCM-R is a spatially explicit model, the results allow mapping of the rank order of restoration potential, providing watershed managers an overall view of where restoration efforts are needed first. Finally, the model allows watershed managers to examine multiple restoration strategies to determine the most effective approach for improving a riparian zone’s score. Future applications could include economic benefit/cost analysis, such as those recently conducted by Daigneault, Eppink & Lee (2017), in the WQCM-R rankings. This addition would allow further prioritization of restoration efforts, focusing on those that provide the highest benefit/cost potential as well as ecosystem services improvement, and provide incremental analysis insights when watershed managers are working with limited budgets under improved water quality goals.

The authors would like to recognize Dr. Bruce Hunter’s and Ms. April English’s unique contributions to the development of the original modeling effort that prioritizes sub-watersheds in terms of preserving high quality riparian zones and the underlying foundation that those efforts had in the development of the model focused on restoration reported here.

Additional Information and Declarations

Competing Interests

Author Contributions

Data Availability

1 NHD stream hydrography and sub-watershed topology is 99.98% identical to NHDPlus topology for the Lewisville Lake watershed (8,615 segments out of 8,617 segments were identical). Because NHD data were used in the original WQCM-P model, we continued to use the same dataset for WQCM-R.

The authors declare there are no competing interests.

Samuel F. Atkinson conceived and designed the experiments, performed the experiments, analyzed the data, contributed reagents/materials/analysis tools, prepared figures and/or tables, authored or reviewed drafts of the paper, approved the final draft.

Matthew C. Lake performed the experiments, analyzed the data, contributed reagents/materials/analysis tools, prepared figures and/or tables, approved the final draft, mr. Lake contributed to the GIS analysis.

The following information was supplied regarding data availability:

The data is available at the UNT Data Repository: Atkinson, Samuel F. Water Quality Corridor Management for Restoration (WQCM-R) Modeling Dataset, dataset, June 10, 2019; https://digital.library.unt.edu/ark:/67531/metadc1506243.

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
