# Peer review of "Prioritizing riparian corridors for ecosystem restoration in urbanizing watersheds"

_PeerJ, doi:10.7717/peerj.8174_

## Round 0.1 · original submission · Major Revisions

Based on the reviewer's comments this is flagged as requiring a major revision but it's really partway between major and minor.

Two comprehensive reviews have been provided and they raise a series of relevant questions. One additional point is that the abstract may be interpreted as restoration being preferred to preservation. Even though the paper does not say this (the previous model was for preservation), it woul be good to remove this ambiguity and say a word or two about their complementary nature and where each is most appropriate.

Reviewer 1 ·

Basic reporting

Riparian restoration is an important subject for the importance it has in the ecosystem integrity preservation and the maintenance of ecosystem services. The manuscript is well written and deals with a methodology to identify sites in need of riparian restoration. It is not a very novel subject as there are many papers already dealing with it, but still in need of further research. One possible limitation is being done at watershed level and not at corridor level, at least in this manuscript. Also, it is not clear the relative expression of human pressures, notably the cited urbanization, at a proper special scale. Results and Methods have mixed parts, Discussion and Methods have mixed parts, and they should be separated. In fact, there is no discussion if you take out the methods and results in it.

My first concern is about the so-called urbanization and the other human alterations affecting the corridors. You do not provide information on the urbanization degree of the area. Please provide a map/quantification/table of the degree of urbanization and other land use in your watersheds and in the corridors that you sorted from the watersheds.
We have no details of the watersheds enough to understand if they are very different.

Experimental design

Background and others. You do not need to say (for purposes of anonymity, citation will be added later). Just do not cite.
A spatially explicit methodology probably should not be called a model. Maybe you would consider to change the name, e.g. to methodology, frame, assessment system or strategy?
Literature review is good, including relevant papers. However, you should avoid grey references when it is not needed or else indicate the site where it can be found, e.g. Atkinson et al 2007. Also some references are incomplete, e.g. Feld et al 2018.
You present the results of PCA without having describe it in the methods and give details.
You present the results of LRA without having describe it in the methods and give details. The description in lines 290-293 are obvious to the model, you do not need to put them here, but you need to describe statistical details, notably transformation of variables, and the independent variable, also the spatial scale used.
Figure 1 is already published and this should be avoided, you can just cite it. The Figure you mention in line 146 can use the present Figure 2.
Figure 3 is too simple, it does not add to the subject in study itself and you do not need it, erase it. Figures 4 is totally inadequate in my view. Instead of bars you should use points/dots. You can make a combined Figure using Figures 4 and 5. Figure 5 shows poorly the results using bars and is highly related to the previous. Similar considerations for Figure 6, it can be included in a single Figure, with different dot types.
You should combine Tables 1,2,3 and 4 in a single Table.
The Discussion includes methods (first paragraph), it simply cannot be done like this. Put the simulations on the methods, show the results, and then discuss here.

Validity of the findings

Basically you make a 20 m buffer, and overlay it with Land Use (it is not a vegetation parameter but a land use parameter, you should change it), erodibility and slope (uphill and not river). So you are mixing human pressures with geophysical characteristics, to deploy a gradient of change. This means that the results will be driven in principle by human uses, in other words, the more human pressure, the more restoration. But can some of these patterns be related to other aspects, such as geological constrains, river slope or others? This can be seen if you make a regression between WQCM-P and R values, and if they are negatively related. Have you done so? If this is the case, why not use the opposite of WQCM-P, including the FEMA, and be developing a WQCM-R?
In fact, the floodplain in particular should be restored, and I do not agree in taking out the 100-year floodplain from the restoration assessment. You should present the possibility of using WQCM-inverse for the restoration identification, and discuss it together with your present proposal, notably when the simulations of different restorations are done.
If the regression model is so good (lines 295-298) why do we need the WQMC-R model an use the regression equations instead? Can you address this in the discussion?

Reviewer 2 ·

Basic reporting

This paper uses a GIS modeling approach to identify areas that are importance for conservation to protect water quality. The writing is clear and the workflow is sufficient for the intentions of the project.

As is, this manuscript describes the modeling methods, but does not sufficiently emphasize the "so what" for the analysis region. What do the results mean? The authors show them, but do not provide much discussion of what the results mean on the ground for managers and the public. It appears that figures one and three show how areas in need of restoration are those that were highly degraded and therefore not conservation priorities in the past. This should be more clearly emphasized in a local context rather than the broader discussion used in the introduction and conclusions.

As is, the framing is clearly based on water quality, but the tool is for riparian restoration priority. Riparian areas may protect water quality, but this is not their only value. Please reframe the introduction more regionally toward the services relevant in northern Texas. Please see the work of Fremier et al on using riparian corridors as conservation corridors: https://www.sciencedirect.com/science/article/abs/pii/S0006320715002529

Experimental design

The research is clear in its workflow and rationale, and falls within the scope of the journal. It is a novel application of a modeling framework to inform watershed management and water quality. However, the methods make trade-offs between established alternative data and methods, and this should be mentioned. Don't just say what you did in the materials and methods, but why, and in some case why other methods weren't selected.

There should be a clear rationale for why a 20-meter buffer was used instead of a threshold tied to the active floodplain. For example, there are several tools that use the 100-year floodplain and then proceed with GIS analyses based on the valley bottom width or a similar high magnitude flood extent. While I don't expect the authors to reproduce their analyses at this extent, they should write a paragraph acknowledging the trade-off they make between studies that perform similar watershed analyses based on valley bottoms. Anytime an arbitrary threshold is picked for a large landscape, this should be justified based on the fluvial geomorphic and hydraulic context it provides to a riparian zone. For example, in some canyons or gorges, the 100-year floodplain might be narrower than 20-meters, while in larger wadeable streams and rivers' unconfined valley bottoms, it may be much larger.

Macfarlane 2015
http://www.sciencedirect.com/science/article/pii/S0169555X15302166
Macfarlane et al. 2016 http://www.sciencedirect.com/science/article/pii/S0301479716308489
O'Brien 2019 https://onlinelibrary.wiley.com/doi/abs/10.1002/esp.4615
Gilbert 2016: https://www.sciencedirect.com/science/article/pii/S0098300416301935
Nagel et al 2014: https://www.fs.fed.us/rm/boise/AWAE/projects/valley_confinement.shtml

Several of these studies can/should be acknowledged as examples of landscape scale riparian planning tools.

Given that a 30-meter DEM was used for the topography, it seems like the buffer width may be small for many landscapes, so once again acknowledge this and justify your approach in the methods. Does this coarse resolution slope data work well in the topography of north Texas, which is flatter than mountainous watersheds with more relief.

Why was NHD used instead of NHD+ or NHD+HR? This is a fair thing to do, but should be mentioned. https://www.usgs.gov/core-science-systems/ngp/national-hydrography

Why was vegetation based on eight classes derived from classified LANDSAT imagery instead of just using the LANDFIRE product provided by USGS/USFS? The existing vegetation height and existing vegetation type parameters may be quite useful here:

https://www.landfire.gov/

Validity of the findings

The findings appear to be correct although the presentation would be more useful if specific conservation priorities were discussed in the context of the study area.

Additional comments

The WQCM-R model is basically the inverse of their earlier WQCM model, which begs the question, why not just assume areas of low conservation priority are degraded and likely need restored? You should really discuss how the models complement one another or whether one model run provides both results by deductive logic. Also - use the continuous and categorical priority values for the figures, not just the categorical values. Add a scale bar and north arrow to any map, labeling nearby geographic landmarks.

---

## Round 0.2 · Minor Revisions

The manuscript has been accepted by reviewers but one has some points about the expression in the paper (style). To give you an opportunity to incorporate their comments, should you wish, I am returnig it with a minor revisions decision.

Reviewer 2 ·

Basic reporting

I appreciate the authors' attention to the comments and review feedback they received. Several changes have been made and the manuscript is in better shape than when it was first reviewed. Regretfully, the manuscript is still not of sufficient writing quality. At a bare minimum, I encourage the authors to revise their abstract, introduction, and conclusions to make sure that all words are necessary and that each sentence is concise and to the point. My attention to the quality of writing is not to punish the authors, but to make sure that their hard work does not go unread, and uncited. Strong verbs and clear adjectives and nouns would greatly improve many passages. I have made a few line edit suggestions, but cannot apply similar changes across the document, as this is not the point of peer review.

By cleaning up the writing, this manuscript will be more attractive to parties looking for an example of a GIS-based floodplain restoration prioritization framework. The references have been improved, but are thin for the length of the manuscript and given the large body of GIS-based floodplain tools that this is similar to.

The conclusions still read more like a methods paper than the "so what?" I would suggest. However, this is at the discretion of the authors.

Experimental design

The design is of sufficient quality, but the writing should be reread and revised as appropriate. There are many ways to skin a cat, and some might disagree with the 20m buffering on each side of the stream based on 30m topography, as well as how riparian/floodplain areas are defined, but I think the recent revisions explain the rationale enough to be useful. The authors performed their due diligence on the NHD request, which is greatly appreciated (in mountainous terrain, the networks can differ substantially, so hopefully it wasn't perceived as a wild goose chase).

Validity of the findings

The findings are valid, although, as cautioned in the first review, the interpretation and applications are what will make this contribution stand out.

Annotated reviews are not available for download in order to protect the identity of reviewers who chose to remain anonymous.

---

## Round 0.3 · accepted · Accept

Thanks for doing the further editing. Paper looks great and I hope will get people really thinking about how complex an area this is, but too important to put in the too hard basket.